# The Landscape of Global Ocean Microbiome: From Bacterioplankton to Biofilms

**DOI:** 10.3390/ijms24076491

**Published:** 2023-03-30

**Authors:** Jie Lu, Yi Shu, Heng Zhang, Shangxian Zhang, Chengrui Zhu, Wei Ding, Weipeng Zhang

**Affiliations:** 1Institute of Evolution & Marine Biodiversity, Ocean University of China, Qingdao 266100, China; 2MOE Key Laboratory of Marine Genetics and Breeding, Ocean University of China, Qingdao 266100, China; 15254135903@163.com; 3College of Marine Life Sciences, Ocean University of China, Qingdao 266100, China; 4Haide College, Ocean University of China, Qingdao 266100, China

**Keywords:** global ocean microbiome, metagenomics, marine biofilms

## Abstract

The development of metagenomics has opened up a new era in the study of marine microbiota, which play important roles in biogeochemical cycles. In recent years, the global ocean sampling expeditions have spurred this research field toward a deeper understanding of the microbial diversities and functions spanning various lifestyles, planktonic (free-living) or sessile (biofilm-associated). In this review, we deliver a comprehensive summary of marine microbiome datasets generated in global ocean expeditions conducted over the last 20 years, including the Sorcerer II GOS Expedition, the *Tara* Oceans project, the bioGEOTRACES project, the Micro B3 project, the Bio-GO-SHIP project, and the Marine Biofilms. These datasets have revealed unprecedented insights into the microscopic life in our oceans and led to the publication of world-leading research. We also note the progress of metatranscriptomics and metaproteomics, which are confined to local marine microbiota. Furthermore, approaches to transforming the global ocean microbiome datasets are highlighted, and the state-of-the-art techniques that can be combined with data analyses, which can present fresh perspectives on marine molecular ecology and microbiology, are proposed.

## 1. Introduction

The history of metagenomics can be dated back to the report of the “great plate count anomaly”, which evidenced that cultured microorganisms cannot represent the whole microorganism world [1]. Then, after the notion emerged that 16S rRNA genes can be used as a marker for differentiating between bacterial species, environmental DNA was directly cloned from the ocean for the 16S rRNA gene sequence analysis [2]. The first definition of metagenomics was known as the “function-based analysis of the mixed environmental DNA species” [3]. Metagenomics studies the total DNA in a specific environment rather than the total DNA in a specific microorganism or its cells, and it avoids the traditional methods of microbial isolation and culture to directly extract total DNA from environmental samples. It does not require the isolation, culture, and purification of microorganisms, providing an alternative way for us to understand and utilize more than 95% of uncultured microorganisms [4,5,6,7,8]. It is a new idea and a new method for studying microbial diversity and developing physiological active substances (or acquiring new genes) on the basis of microbial genomics. The term “metagenomics” was redefined after applying whole genome shotgun sequence (WGS) technologies to microbiota for both taxonomic and functional analyses [9,10]. Technically, the steps of metagenomics can be divided into four parts: environment samples collection, DNA extraction, WGS sequencing, and analysis. Since then, metagenomics has developed rapidly as advanced, high-throughput sequencing technology made the rapid acquisition of massive microbiome datasets possible [11].

Metagenomics research has permeated every field, from human bodies to animals and from land to the marine environment [12]. Marine microbiota, referred to as microbial communities living in marine environments, are estimated to comprise approximately four gigatons of carbon [13]. Their unique metabolisms allow marine microbes to play an essential role in biogeochemical events, represented by the carbon cycle [14,15], nitrogen cycle [16,17,18,19], and sulfur cycle [20,21,22]. Marine microbiota can survive in different marine environments, and some can even survive in oligotrophic conditions [23,24], low-temperature conditions [25,26,27], high-pressure [26,28,29], high-temperature [30,31], high-salt [26,32] and other extreme environments [33] and can continue to produce offspring. Thus, the study of marine microbial diversity can not only further our understanding of the interactions between microbes and environments but also provide an important reference for the development and utilization of marine microbial resources. However, most microbial research is limited to microbial isolation, purification, and cultivation in the laboratory [34,35]. Therefore, the fact that many marine microbes remain uncultured has complicated and hindered marine microbiology and ecology studies for a long time. This has greatly limited our understanding of the microbial world. By applying metagenomics to study the whole sets of population genomes from microbiota, the critical role of marine microbiota can be better understood. A number of metagenomics studies, including several global-scale projects, have been conducted to explore the identities and functional properties of marine microbiota, and a relevant “data explosion” is being witnessed [36,37,38]. Moreover, the distribution pattern of any functional gene or species can be compared between metagenomes from different environments and projects [36,37,38]. Global comparisons will help identify ecosystem specificities, such as microbial participation in and in response to climate change (e.g., carbon, sulfur, and nitrogen cycles) [39,40,41,42], human health risks (e.g., the presence of pathogen species, toxin genes, and viruses) [43,44,45,46,47], and biodegradability [48].

Marine microbes have three major lifestyles: the planktonic lifestyle (free-living), sessile lifestyle (biofilm-associated), and symbiosis (e.g., intracellular bacterial symbionts). A biofilm forms when marine microbes attach to substrates immersed in seawater, such as rock surfaces in the subtidal zone [46], hydrothermal vent chimneys in the deep oceans [49,50], marine animal body surfaces [51,52], organic particles within the marine sediments [53,54], ship bottoms [55], and micro-plastics [54,56], which are causing increasing environmental problems. Thus, biofilms are distributed nearly everywhere in the global ocean. According to an estimation, biofilms constitute 40% of the ocean microbial biomass [57]. A marine biofilm community is usually composed of hundreds and even thousands of microbial species, reaching higher diversity and richness than the adjacent seawater microbiota [36]. In terms of physiological structure, biofilms are characterized by a heterogeneity of stratified nutrients, oxygen concentration, and signaling molecules [58]. Moreover, marine biofilms are reported to be involved in important biogeochemical processes, such as converting dissolved organic matter [59], re-mineralizing particles [60], and passing nutrients from surface waters to deep oceans [61]. Compared to the free-living microbiota and symbionts, studies on marine biofilms have barely begun, likely due to their complex physical community structure and taxonomic composition. Using metagenomics, our studies [36,62,63,64] explored the developmental process of marine biofilms (Figure 1). These studies further support the idea that advances in marine system ecology can be greatly facilitated by the increasing availability of metagenomic data, which provide information on microbial communities in a particular place and time. In this review, we summarized the datasets generated in the global ocean metagenomics studies, including the information on microbiota living in seawater and marine biofilms. We also stated the global ocean datasets beyond metagenomics and stated how to transform metagenomics datasets into ecological or biological discoveries. Finally, we proposed bottlenecks in the development of metagenomics studies and state-of-the-art techniques (e.g., deep learning) that can be integrated into metagenomics studies toward the deep mining of useful information.

## 2. What Ocean Environments Have Been Analyzed

The metagenomics information generated by representative the global ocean sampling projects is displayed in Figure 2. Launched in the year 2004, the Sorcerer II Global Ocean Sampling (GOS) Expedition traversed across thousands of kilometers from the North Atlantic to the South Pacific and collected surface seawater, including a total of 41 different samples [65,66]. An extensive dataset of 7.7 million sequenced reads (6.3 billion bp) was obtained through whole-genome shotgun sequencing to address questions related to genetic and biochemical microbial diversity [65,66]. The major finding of the GOS metagenomics study included that microbial diversity is largely organized around phylogenetically related but geographically dispersed populations [65,66]. There is considerable variation among these microbial populations, both in variable genomic sequences and in hypervariable genomic islands, and environmental-factor-based specific genetic differences can be detected [65,66]. For instance, the gene family of proteorhodopsin is likely to contribute to energy metabolism under low-nutrient conditions [65,66]. In addition, using a novel clustering method based on full-length protein sequence similarity, it was suggested that the GOS dataset almost covered all known microbial protein families, and this method also revealed a great number of unknown protein families [65,66]. These datasets and analyses have laid the foundation for expanding our understanding of individual microbial lineages and their evolution, the organization of marine microbial communities, as well as how they are influenced by the environment.

Initiated in September 2009, the *Tara* Oceans project collected 243 seawater samples of upper and mid-layer seawater from 68 sites in the world’s oceans for metagenomic analysis, resulting in 7.2 TB of datasets [37]. This project has generated resources represented by the ocean microbial reference gene catalog (OM-RGC), which includes over 40 million nonredundant sequences from viruses, prokaryotes, and picoeukaryotes [37]. Novel ecological findings have been made based on these datasets, such as the enrichment of aerobic respiration genes in the mesopelagic compared with the epipelagic zone, implying that the mesopelagic zone is a key re-mineralization site of exported production [37]. Species diversity and functional richness also increase with depth [37,67], consistent with the previous understanding that diversified species are adapted to a wider range of niches, such as particle-associated micro-environments in the mesopelagic zone [68]. In addition, the *Tara* Oceans datasets revealed that epipelagic community composition is mostly driven by temperature rather than other environmental factors or geography, while the mesopelagic community is mainly influenced by carbon sources [37].

The datasets documented in the bioGEOTRACES project during 2010–2011 presented whole-community metagenomes of 610 samples collected in the Atlantic and Pacific Oceans [69]. These metagenomes contained genomic information from a diverse range of bacteria, archaea, eukaryotes and viruses, providing snapshots of microbial communities across space and time that were associated with physical and chemical measurements. Then, the Marine Microbial Biodiversity, Bioinformatics, and Biotechnology (Micro B3) project investigated global ocean microbial biodiversity and collected samples on a single orchestrated day (21st June 2014), aiming to generate the largest standardized microbial dataset [70,71]. This dataset is a powerful source of motivation, since it can pave the way for prioritizing scientific research and raising public awareness about the invisible majority of the world’s oceans [70,71].

More recently, the Bio-GO-SHIP project presented 971 globally distributed surface ocean metagenomes, collected at a high spatio-temporal resolution with a median distance between sampling stations of 26 km [72]. One of the primary goals of the GO-SHIP is to make high spatial and vertical resolution measurements of key state variables to directly quantify the impact of climate change on the marine environment, which can help answer questions about the link between microbial communities and biogeochemical fusion in a changing ocean [72]. In addition, most Bio-GO-SHIP samples are collected every 4–6 h so that daily changes in microbial composition and gene content can be analyzed [73]. Data from this project also suggest the need for increased sampling of marine metagenomes in the central and western Pacific above 50° N and 50° S and below the light transmission zone [72,73].

Yet the microbiomes at the Earth poles have not been well-documented in the above-mentioned projects, which motivated us to correct such a flaw by collecting 60 seawater samples at 28 sites at multiple depths in the Arctic and the Antarctic [38]. The samples underwent metagenomic sequencing and were compared with data from the *Tara* Oceans project [38]. A systematic comparison between the polar and non-polar metagenomes suggested that the occurrence of dominant and locally enriched microbes in the polar seawater with unique functional traits for environmental adaption and the Arctic and Antarctic microbiomes are more similar to each other at the functional level than from a taxonomic perspective [38]. The metabolic pathway reconstruction of these microorganisms demonstrates versatility in sugar and lipid biosynthesis, nitrate and sulfate reduction, and carbon dioxide fixation [38]. A comparison of Arctic and Antarctic microbiota shows that antibiotic resistance genes are enriched in the Arctic, while functions such as DNA recombination are enriched in the Antarctic [38]. This study also constructed the first functional gene catalog of the polar marine microbiota. The main functional basis of the polar microbiome is associated with low-temperature adaptation and environmental change [38].

However, the studies mentioned above are confined to the free-living microbes, omitting microbiota associated with surfaces immersed in the seawater. In a study published in 2019, we profiled the taxonomic and functional structure of 101 marine biofilms that were collected across three ocean provinces during 2011–2019 [36]. A comparison between these biofilms and the surface seawater metagenomic data of the *Tara* Oceans project revealed over 7300 new species, increasing the known microbial diversity in the ocean by more than 20% [36]. It provides evidence for the differentiation of marine ecological niche, and further supports the underestimation of marine microbial diversity [36]. Moreover, a biofilm-specific gene catalog has been created which consists of 195,639 non-redundant protein-coding genes, and a biofilm core has been proposed that comprises functions of stress responses and microbe–microbe interactions [36]. An analysis of 479 genomes reconstructed from the biofilm metagenomes revealed novel biosynthetic gene clusters and CRISPR-Cas systems [36].

## 3. Global Ocean Datasets beyond Metagenomes

While metagenomics reveals the functional potential, meaning what the microbes have the ability to do, metatranscriptomics and metaproteomics indicate the functional processes that are happening [74]. To a large extent, the results derived from the latter two methodologies are more “right” and come closer to truth. However, for reasons related to a high cost and sampling approach, datasets across the global oceans have long been restricted to metagenomics. The combination of metagenomic and metatranscriptomic data to quantify gene expression levels, such as the number of expressed transcripts per gene, has revealed a number of important insights. For instance, Picocyanobacteria have been found to contribute more to the community pool of transcripts than expected based on metagenomics abundances, whereas some heterotrophic bacteria, including those from the highly abundant SAR11 clade, have been found to have done the opposite [75]. A pool of community transcript (metatranscriptome) changes from one sample to another can also be studied by metatranscriptomics in addition to differences in gene and transcript abundances within samples. For example, a metatranscriptomic survey of microbial community metabolism in oxygen minimum zones (OMZs) revealed gene-specific expression patterns and identified key functional groups for taxon-specific genomic profiling [76]. Moreover, an article [77] published in 2019 reported 187 metatranscriptomes from 126 globally distributed sampling stations and established a resource of 47 million genes to study community-level transcriptomes across depth layers from pole to pole and then drew the respective gene expression profiles. By analyzing the correlation between the axes of environmental variations and gene expression changes, an ecological principle governing the community dynamics and gene expression turnover was hypothesized: in polar regions, the relative contribution of gene expression regulation to the response of global warming is lower than that of microbial composition shift; however, an opposite response is shown in the non-polar regions [77].

In comparison with metatranscriptomics, metaproteomics datasets across global oceans are even rarer or have not been documented yet, and there are only some metaproteomics studies that are restricted to a few sampling locations. Using comparative membrane metaproteomics, researchers discovered functional responses of microbial communities to different nutrient concentrations on an oceanic scale, revealing changes in nutrient utilization and energy transduction along an environmental gradient across the South Atlantic waters [78]. Key microbial players driving carbon and nutrient cycling in a seasonally stratified fjord were mapped onto in situ metabolic networks, using a combination of metaproteomic and metagenomic methods in OMZs [79]. The spatial and temporal patterns of gene expression for nitrification, anaerobic ammonium oxidation (anammox), denitrification, and inorganic carbon fixation differed across the redoxcline and correlated with the distribution patterns of ubiquitous OMZ microbes. In a metaproteomics work published in 2016 [80], we reported the community-wide post-translational modifications (PTMs) in a hydrothermal vent biofilm dwelling in the South Mid Atlantic Ridge and the PTMs enriched in genes that participate in energy metabolism, signal transduction, and inorganic ion transport. This work also found methylation to be one of the main PTM events, while phosphorylation was a rare type.

## 4. Transformation of the Large Datasets

To a significant extent, these large datasets are simply superficial profiles until people start to dissect and digest them. For example, through searching against publicly available metagenomes, the distribution of *hgcAB* genes for a new global view of Hg methylation potential has been profiled: they were abundant in nearly all anaerobic environments, including the oxygenated layers of the open ocean [81]. Similarly, the distribution characteristics of genes corresponding to many aspects of ecological or biological functions in marine environments have been continuously investigated, such as the *mac* gene cluster, which mediates interaction between biofilms and the marine fouling organism *Hydroides elegans* [82], antibiotic resistance genes [83], the *sox* gene cluster, which is responsible for thiosulfate oxidization [84], the microbial terpenoid lipid cyclase, which interprets ancient microbial communities [85], the quorum-sensing genes involved in biofilm development [86], the mobile genes contributing to microbial gene exchange and evolution [87], and the DNA methylation patterns associated with virus–host dynamics in the ocean [88].

Mining these large datasets with a focus on a specific function not only reveals the genes’ distribution and diversity but also facilitates the discovery of new genes or new functions of known genes. For example, based on GOS metagenomic sequences, a great number of protein families, including phosphatases, proteases, glutamine synthetase, and ribulose-1,5-bisphosphate carboxylase/oxygenase (Rubisco), were discovered [65,66]. Through integrating metagenomic analyses and biochemical approaches, a form I Rubisco that lacks small subunits was discovered [65,66]. Moreover, tools for microbial research, such as the 16S rRNA primers used for identifying a microbial community structure, can be significantly improved using metagenomes from global ocean surveys [89]. There has been a long-standing debate regarding the accuracy of different conserved regions of the 16S rRNA gene in determining the taxonomy of microbes, depending on the consistence between the primer sequence and the microbial genes in the target microbiomes. To solve this problem, analyzing the global ocean metagenome can easily reveal which sequence region is more conserved among most of the microbes [89].

Furthermore, the strategy driven by metagenomics can study enzymes and natural products not previously described in the under-developed microbiota and environment. In 2021, a systematic and integrated study on the Earth’s microbiomes [90] emphasized the value of the genome-centered approach in revealing the genome property of uncultured microorganisms that affect ecosystem processes. They assembled and boxed 10,450 globally distributed metagenomes from different habitats, including marine and other aquatic environments (N = 3345), human and animal host-related environments (N = 3536), and soil and other terrestrial environments (N = 1919) to recover 52,515 metagenome-assembled genomes (MAGs) [90]. This study not only expanded the known phylogenetic diversity of bacteria and archaea by 44% but also proved the practicability of this collection in understanding the biosynthetic potential of secondary metabolites and solving the connection between thousands of new hosts of uncultured viruses [90].

In a more recent study [91], the biosynthetic potential of the global ocean microbiome was deeply studied through analyzing approximately 35,000 microbial genomes derived from single cells, cultivated microbes, and MAGs from seawater metagenomes with the aim of transforming the large datasets into gene resources. It turned out that tropical and surface communities are the most promising sources of new terpenes, while the least explored communities (polar, deep, and those rich in viruses and particles) have the greatest potential in discovering NRPS, PKS, RiPPs and other natural products [91]. Representative approaches to transforming the global ocean datasets are shown in Figure 3.

## 5. Bottlenecks and Future Directions

Despite the increasing number of the global ocean omics studies, there are still many types of ocean microbiota that are barely studied, such as the gut microbiota of animals, especially those living in the deep sea [92,93,94]. Here, we would like to highlight a recent study [95] in which more than 5000 genomes comprising the intestinal microflora of 180 wild animals living on land were analyzed, including different species, feeding behavior, geographical locations, and characteristics. The authors determined that the intestinal microflora of wild animals is a largely untapped resource and proved that it is a promising resource for discovering new biological functions and technologies [95]. This study also discovered that microbial composition, diversity, and function are related to animal classification, diet, activity, social structure, and life span [95]. Compared to the gut microbiota of animals living on land, the genomic information and functional properties of microbes residing in the bodies of marine animals are largely unknown, and those that are well-known are only confined to the endosymbionts. This is due to the notion that endosymbionts are often essential for their animal hosts by providing carbon sources (e.g., chemoautotrophic bacteria fixing carbon dioxide) [96] or oxidizing toxic compounds (e.g., sulfur-oxidizing archaea consuming sulfide gas) [97], and a strict specificity can be observed in regard with the identities of the symbiont and the host. In contrast, gut microbiota in the gastrointestinal tracts of marine animals seems to be easily changed by environmental fluctuations (e.g., a temperature increase) [98,99]. However, there is accumulating evidence to demonstrate the ecological function of marine animal gut microbiota. For example, certain microbes in the scallop gut can detoxify paralytic shellfish toxins, which are a group of potent neurotoxins that cause paralytic shellfish poisoning [100]. Moreover, “changeable” does not necessarily mean “dispensable” because the core members may be still there. A systemic analysis of marine animal gut microbiota, including their genomic functional profiles and the strength of their genetic association with their hosts, will be helpful to resolve the existing controversy.

Another important issue is that there is still a gap between the predicted and experimentally evidenced gene functions. This is partially due to the time-consuming path in validating the function of a target gene using biochemical approaches (e.g., enzymatic assay) and genetic manipulation (e.g., gene knock-out). In actuality, because most of the marine microbes are still uncultivated, there is not even the opportunity to conduct experimental trials. Therefore, strain isolation and cultivation would be the first step in filling the research gap. In the field of research addressing human gut microbiota, extensive efforts have been spent on microbial isolation and genome sequencing and unlocking the principles that govern the heritability of microbiota. For example, the resident bacterial community of the human intestinal tract was determined by combining large-scale whole-genome sequencing and bacterial culture [101]. The phenotypic and genomic analyses show that spore-forming bacteria and non-spore forming bacteria represent the main and different phenotypic components of human microbiota, each with unique colonization dynamics [101]. As important as these are, the isolation of human gut bacterial species has revealed a marked proportion of oxygen-sensitive bacteria that can be transmitted between human individuals by producing spores [101]. However, the cultivation of marine microbes is significantly lagging behind in comparison with those living on land. Therefore, although the development of novel cultivation methodology may be an agent issue, there is still a long way to go.

Finally, continuous advances in cutting-edge technologies, such as those in the scope of artificial intelligence and synthetic biology, will promote crossover studies between marine microbial ecology, computational science, and biology. From one perspective, with the expanding datasets of the global ocean microbiomes, novel and useful gene resources (e.g., antimicrobial peptides) can be discovered by using precise predicting scripts that are developed using deep learning. In addition, the products generated through the global ocean datasets can include novel enzymes such as endoglucanases that metabolize cellulose into sugars for further metabolism to ethanol, such as those discovered through the mining of oil microbiomes [102]. Alternatively, with the cultivated strains, we can apply models of a synthetic microbial community by mixing numerous single strains in the framework of the synthetic biology to test ecological theories. For instance, the effect of global warming on a marine microbial community cannot be easily tested in situ. Moreover, community-level understanding and the optimization of bioprocesses can be facilitated by using the synthetic community model, and these optimized bioprocesses have a wide range of utilization, such as in bioremediation [103]. It is due to these advances that the true value of marine microbiomes is just beginning to be presented.

## Figures and Tables

**Figure 1 ijms-24-06491-f001:**
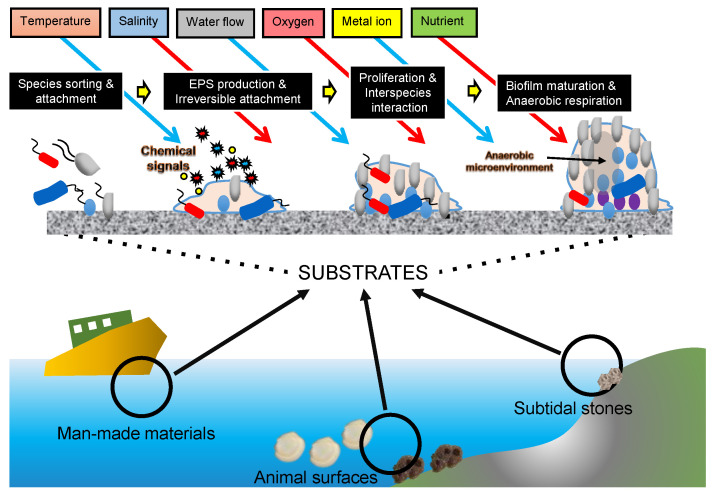
A schematic model of marine biofilm development.

**Figure 2 ijms-24-06491-f002:**
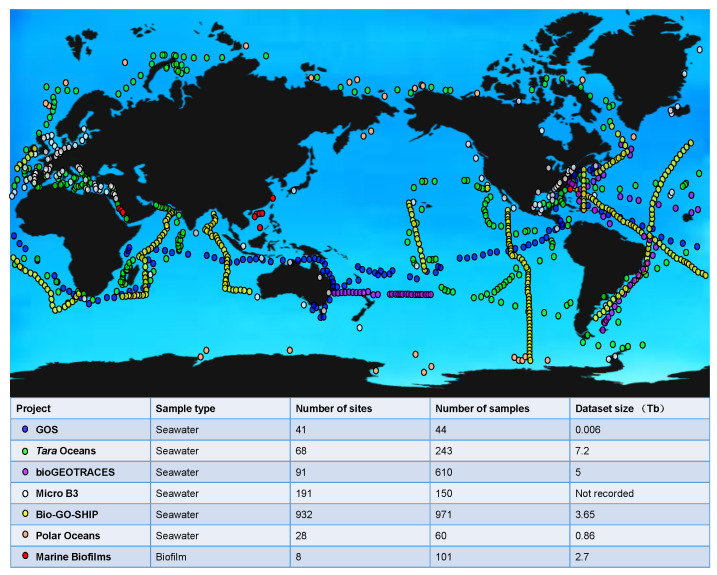
Summary of the global microbiome information. The sampling sites are plotted in the global ocean map. The launch time of these projects [36,37,38,65,66,69,70,71,72,73], number of sampling sites, number of samples, and the dataset size are summarized in the bottom panel.

**Figure 3 ijms-24-06491-f003:**
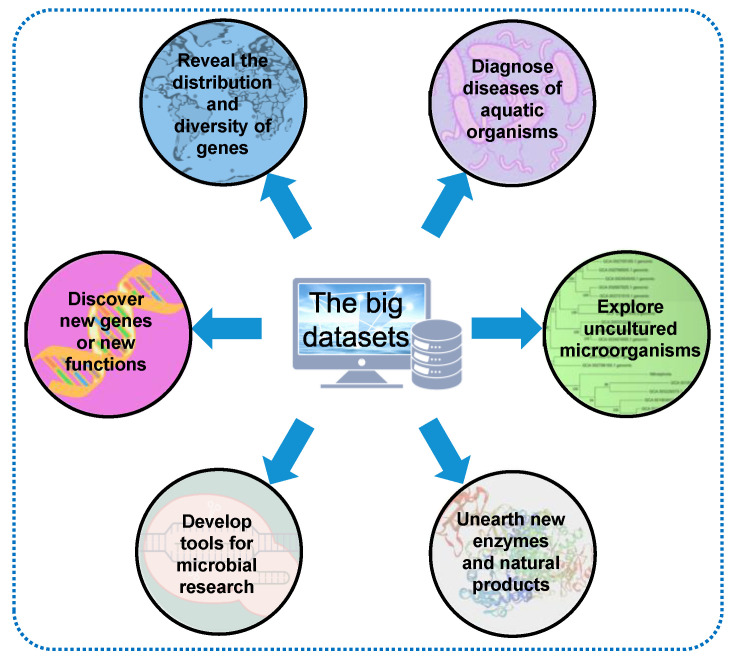
Transformation of the global ocean microbiome datasets.

## Data Availability

Not applicable.

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
