# Peer review of "The Landscape of Global Ocean Microbiome: From Bacterioplankton to Biofilms"

_ijms, 2023, doi:10.3390/ijms24076491_

Round 1
Reviewer 1 Report
Comments
The Review article is well presented. The major issue is what products can be obtained from this resource.
What are the potential applications of the information present in the Marine ecosystem?
Ref.: https://doi.org/10.1007/s12088-022-01040-x.
https://doi.org/10.1007/s12088-019-00842-w
Author Response
Thanks for your positive comments. Here we have revised accordingly. “From one perspective, with the expanding datasets of global ocean microbiomes, novel and useful gene resources (e.g., antimicrobial peptides) can be discovered, by using precise predicting scripts developed by deep learning. Besides, the products generated through the global ocean datasets can include novel enzymes, such as endoglucanases metabolizing cellulose into sugars for further metabolism to ethanol, like those discovered through the mining of oil microbiomes [102]. Alternatively, with the cultivated strains, we can apply models of synthetic microbial community by mixing numerous single strains, in the framework of the synthetic biology, to test ecological theories. For instance, the effect of global warming on a marine microbial communities cannot be easily tested in situ. More-over, community-level understanding and optimization of bioprocesses can be facilitated by using the synthetic community model, and these optimized bioprocess have a wide range of utilization, such as bioremediation [103]. It is because of these advances that the true value of marine microbiomes is just beginning to be presented.”
Reference
- Kalia, V.C.; Gong, C.; Shanmugam, R.; Lee, J. K. Prospecting microbial genomes for biomolecules and their applications. Indian J. Microbiol. 2022, 62(4), 516-523.
- Lee, J. K.; Kalia, V.C. Mapping microbial capacities for bioremediation: genes to genomics. Indian J. Microbiol. 2020, 60(1), 45-53.
Reviewer 2 Report
This mini-review is extremely well-written and covers a wide range of analyzes of microorganisms in the oceans. Given that today there are still not enough studies that deal with the analysis of useful things and benefits of microorganisms of marine origin, research that reveals which groups of microorganisms are found in such ecological systems is important. It is also important to know the biological diversity of the marine environment to better examine the mutual influences of organisms on each other as well as the microorganisms themselves on the ecosystem.
The work is supported by current research, evident from the list of references, and it brings together valuable information in one place.
Author Response
Thanks for your supports.
Round 2
Reviewer 1 Report
Good job